# Do Tabular Foundation Models Learn Rules or Memorize Exemplars?

**Amir Rezaei Balef**[1 2 3]  **Mykhailo Koshil**[1 2]  **Behzad Nourani-Koliji**[3]  **Katharina Eggensperger**[1 2]

## Abstract

We investigate how tabular foundation models (TFMs) generalize during in-context learning (ICL), focusing on whether they rely on memorizing context examples or learning abstract rules. Standard evaluation metrics assess empirical performance, but provide limited insight into the underlying mechanisms driving model predictions. Understanding these mechanisms enables better prediction of model behavior on new downstream tasks and supports more robust applications. To address this, we adapt and extend controlled experimental frameworks to the tabular domain and evaluate six state-of-the-art TFMs under settings that distinguish exemplar-based interpolation from rule-based extrapolation. We empirically examine how generalization behavior evolves with the number of training samples and across model layers. We observe that *models with similar predictive performance can exhibit substantially different generalization strategies, with a common trend of transitioning from exemplar-based behavior in early layers to more rule-based reasoning in deeper layers.* These findings suggest that predictive accuracy alone is insufficient to characterize model behavior.

## 1. Introduction

Tabular Foundation Models (TFMs) have recently emerged as a powerful paradigm for learning from structured data, achieving state-of-the-art performance across a wide range of tasks (Erickson et al., 2025). Despite these empirical successes, current evaluations rely almost exclusively on aggregate performance metrics such as accuracy or AUC,

[1]TU Dortmund University, Dortmund, Germany [2]Lamarr Institute for Machine Learning and Artificial Intelligence, Dortmund, Germany [3]University of Tübingen, Tübingen, Germany. Correspondence to: Amir Rezaei Balef <amir.balef@tu-dortmund.de>.

*Proceedings of the 2^{nd} ICML Workshop on Foundation Models for Structured Data*, Seoul, South Korea. 2026. Copyright 2026 by the author(s).

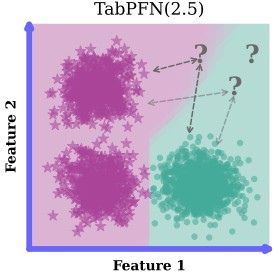 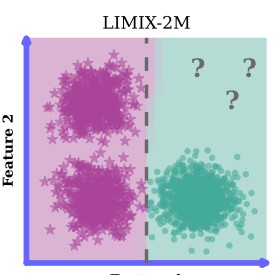

*Figure 1.* An exemplar-based model computes the similarity between test (query) and training (support) examples using all features (more pronounced for *TabPFN(2.5)*). A rule-based model uses a parsimonious decision boundary that explains the data (exhibited by *LimiX-2M*).

which provide limited insight into how these models generalize. In particular, strong predictive performance does not necessarily imply robust or predictable generalization behavior (Geirhos et al., 2020; Oka & Takefuji, 2026).

It remains unclear whether in-context learning in TFMs captures underlying rules or instead exploits similarity between support and query examples. Empirically, as shown in Figure 1, identical training data can induce different predictive behaviors and decision boundaries across TFMs, particularly under out-of-distribution (OOD) conditions. Consequently, this distinction directly affects reliability, robustness to distribution shifts, and extrapolation beyond observed data (Wu et al., 2025).

This work addresses the following central research question:

> **Question.** How do tabular foundation models generalize beyond observed data, and to what extent do they rely on learning underlying abstract rules versus leveraging similarity between training (support) and test (query) samples?

On our path to address this question, we move beyond standard performance metrics to analyze the mechanisms underlying generalization in TFMs, studying six state-of-the-art tabular foundation models. We adapt experimental frameworks (Dasgupta et al., 2022; Chan et al., 2022) to the tabular setting and conduct controlled ablations over support set size. We further examine how generalization behavior evolves across layers. Our results suggest that, despite comparable predictive performance, tabular foundation models

may rely on different generalization strategies, ranging from rule-based to exemplar-based reasoning.

## 2. Related Work and Background

This section reviews prior work on understanding generalization in TFMs with different objectives, as we use this to justify our research target.

Prior work has studied how the ICL abilities of TFMs generalize across different conditions. Nagler (2023); Koshil et al. (2024); Liu & Ye (2025) study how TFMs scale to larger datasets beyond their pre-training settings. Furthermore, Kolberg et al. (2025) investigate how the performance of *TabPFN(v2)* scales with the number of features, while Ye et al. (2025) examine the impact of ensemble size. Zheng et al. (2025) leverage signal reconstruction and frequency response analysis to characterize the inductive biases of *TabPFN(v2)*, showing that its frequency capacity adapts to the number of support samples.

However, these studies primarily evaluate models in fixed-task settings or under controlled shifts such as feature dimensionality or context size (Arbel et al., 2025). In contrast, our work moves beyond performance-centric benchmarks and instead focuses on whether these models rely on deep semantic and causal-like rules (Liu et al., 2026) or whether they primarily function as sophisticated similarity-based retrieval systems (Miftachov et al., 2026) under out-of-distribution (OOD) conditions.

In this regard, it is useful to revisit the underlying mechanism of TFMs. TFMs are transformer-based models pre-trained for supervised learning via ICL, approximating Bayesian inference from labeled examples without weight updates and pre-trained on synthetic prior tasks (Müller et al., 2022). At inference time, support (training) and query (test) samples are processed jointly through a transformer, where self-attention enables the model to condition on relevant examples in the context. This mechanism can be viewed from two complementary perspectives: on the one hand, it implements a learned inference procedure shaped by prior-induced inductive biases (von Oswald et al., 2024; Kirsch et al., 2022). For example, Swelam et al. (2025) show that the embeddings learned by *TabPFN(v2)* encode causal structure that can be reliably recovered; on the other hand, it can be interpreted as a similarity-based retrieval process, where queries attend to relevant examples based on key–query similarity (Pantazopoulos et al., 2026; Dinh et al., 2026).

Building on these perspectives, we rely on the prediction mechanism and its generalization behavior in TFMs using the controlled experimental framework introduced by Dasgupta et al. (2022). In their work, Dasgupta et al. (2022) found that increasing the width of a neural network (NN)

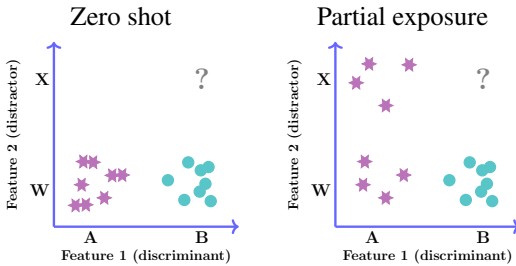

*Figure 2.* Settings used to evaluate model generalization (specified by quadrants): AW, BW, AX for train; BX for test.

with a fixed depth of 1 leads to more exemplar-like behavior. Relatedly, Chan et al. (2022) show that exemplar-based strategies can be advantageous in low-data regimes, where abstract rule formation is difficult, whereas rule-based strategies may improve robustness to spurious correlations in noisy settings.

Building on these findings, we propose a first step towards systematically studying the generalization behavior of TFMs.

## 3. Methodology

To systematically distinguish between rule-based and exemplar-based behaviours, we use the same experimental setup as Dasgupta et al. (2022); Chan et al. (2022), which is based on a 2D binary classification task as shown in Figure 2. The data is organized into four quadrants corresponding to combinations of feature values.

The model is evaluated on the held-out quadrant BX under two settings. The *zero-shot* (zs) setting studies generalization to unseen feature values. The model is given samples from AW (★) and BW (●) and is expected to classify the test samples as green dots (●). The *partial exposure* (pe) setting studies generalization to unseen feature combinations. For this, the model is trained on the same BW samples as in the *zero-shot* setting, while the AW samples are partitioned into the AW and AX quadrants. In this setting, a rule-based model should behave similarly to the *zero-shot* setting, as it relies on abstract rules that generalize across feature combinations. In contrast, an exemplar-based model makes decisions based on similarity to observed examples, causing the additional AX samples to influence its predictions on BX.

To provide a quantitative assessment, (Dasgupta et al., 2022) introduced the Exemplar-vs-Rule (EvR) metric (Dasgupta et al., 2022), which is the performance gap between the *zero-shot* and *partial exposure* settings. This gap measures to what extent a model's behaviour is impacted by the *distractor* features (in *AX*). Formally, for a classifier family

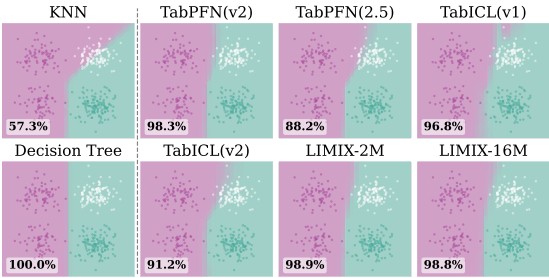

*Figure 3.* Generalization under partial exposure test. Performance under partial exposure is measured on the held-out BX points, where white dots should be classified as green. Comparing KNN as an exemplar-based model and decision trees as rule-based models with tabular foundation models shows that some TFMs tend to be more exemplar-based, while others are more rule-based.

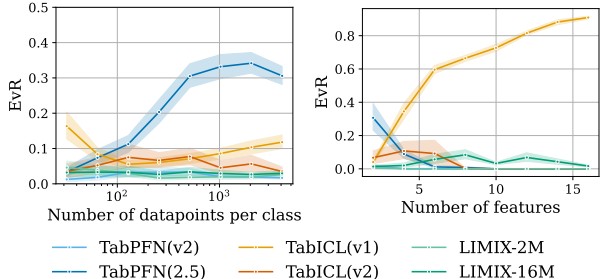

*Figure 4.* **Effect of number of samples and features on generalization.** *TabPFN(2.5)*, *TabICL(v1)* and *TabICL(v2)* change their EvR behavior with increasing number of datapoints or features per class, showing that some TFMs exhibit hybrid generalization.

$\mathcal{F}$,

$$\text{EvR}(\mathcal{F}) = \mathbb{E}_x\big[A(y, f^{\text{zs}}(x))\big] - \mathbb{E}_x\big[A(y, f^{\text{pe}}(x))\big],$$

where $f^* \in \mathcal{F}$ is trained on the corresponding setting and $x$ is sampled from the test quadrant. Notably, the data in each quadrant is generated using a Gaussian random generator (see Appendix A.1 for details). Larger EvR values indicate a stronger reliance on exemplar-based generalization, while smaller values suggest rule-based behavior.

## 4. Experiments

Next, we study TFMs in this framework. In particular, we study two state-of-the-art open-source tabular in-context learning (ICL) models, *TabICL(v1)* (Qu et al., 2024) and *TabICL(v2)* (Qu et al., 2026), as well as four open-weight models, *TabPFN(v2)* (Hollmann et al., 2025), *TabPFN(2.5)* (Grinsztajn et al., 2025), *LimiX-2M*, and *LimiX-16M* (LimiXTeam, 2025). We run all models in their default configurations and their respective data preprocessing pipeline.

For the first experiment, we compute EvR in the exact setting introduced by Dasgupta et al. (2022). In addition to the aforementioned TFMs, we include K-Nearest Neighbors (KNN) and a decision tree as classical machine learning classifiers with known generalization behaviour. As shown in Figure 3, KNN exhibits pure exemplar-based behavior by relying on similarity to training instances, whereas decision trees provide a clear example of rule-based generalization through axis-aligned decision boundaries. TFMs, in contrast, can exhibit qualitatively different decision strategies, ranging from exemplar-like behavior (e.g. *TabPFN(2.5)*) to more rule-based behavior (e.g., *LimiX* model family).

> **Takeaway 1.** TFMs exhibit both rule-based and exemplar-based generalization.

Notably, Dasgupta et al. (2022) argue that both rule-based

and exemplar-based extrapolation can be useful depending on the domain. Given the heterogeneity of tabular data, it is therefore important to understand whether TFMs can *adapt* their behavior across different datasets. Furthermore, Wu et al. (2025) propose hybrid models combining elements of both rule-based and similarity-based generalization, enabling flexible modeling of generalization across different contexts. Motivated by these insights, we design the following experiments to evaluate whether the generalization behavior of TFMs adapts across settings or remains fixed, as quantified by shifts in the EvR metric.

**Effect of the number of training samples on EvR.** Here, we analyze how generalization behavior varies with the number of datapoints per class. In Figure 4 (left), we observe that different models exhibit distinct trends in their Exemplar-vs-Rule (EvR) bias as data increases. While *TabPFN(v2)* and the *Limix* models remain largely stable with relatively low EvR, *TabPFN(2.5)* and the *TabICL* models show a clear dependence on the number of samples. Most prominently, *TabPFN(2.5)* exhibits lower EvR (more rule-based behavior) with few samples and as the amount of data increases it shifts toward higher EvR, indicating greater reliance on exemplar-based generalization. However, with even more samples, the EvR decreases again. This dependence is less pronounced for *TabICL(v1)* and *TabICL(v2)*. Overall, these results indicate that while some TFMs adapt their generalization strategy depending on data availability, others exhibit fixed inductive biases that are less sensitive to sample size. Notably, for this expriment, we report the average EvR across different settings obtained by varying label and feature orderings (see Appendix A.3).

**Effect of the number of features on EvR.** In Figure 4 (right), we fix the number of datapoints per class to 1024, and we extend the experiment to higher-dimensional settings by increasing the number of both distractor and discriminator features. While *TabPFN(v2)* maintains low EvR, other models are more sensitive. As seen, for *TabICL(v1)*, the EvR increases with the number of features, meaning that

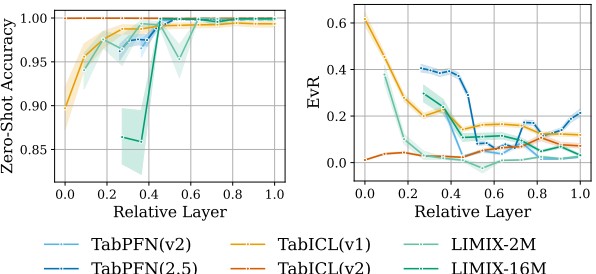

*Figure 5.* **Layer-wise generalization behavior.** Models transition from exemplar-based (high EvR) in early layers to rule-based (low EvR) in deeper layers.

*TabICL(v1)* increasingly fails to capture the underlying rule. However, for the other models, the trend is the opposite with a decreasing EvR with higher dimensions.

> **Takeaway 2.** TFMs may adapt their generalization strategy depending on data availability.

**How does EvR evolve across the internal layers of TFMs?** Here, we analyze how generalization behavior evolves across model depth by computing EvR at different relative layers during inference. To obtain intermediate predictions, we use an adapted version of the "logit lens" method (nostalgebraist, 2020). Following Balef et al. (2026), we further pre-train a separate decoder for each layer on synthetic datasets generated using *TabICL(v1)* priors. At inference time, we extract the hidden representations after each layer and pass them through their corresponding layer-specific decoders to produce intermediate predictions.

Since intermediate predictions from very early layers can be unreliable, as shown in earlier works (Küken et al., 2025; Balef et al., 2025; 2026), for each model, we restrict our analysis to layers that achieve at least $90\%$ of the final performance of the model on the *TabArena* and *PMLBmini* benchmarks. This ensures that only layers exhibiting sufficiently reliable predictive behavior are considered (see Appendix A for details). Notably, in this experiment, we fix the number of datapoints per class to $1024$ and report the average EvR across different settings obtained by varying label and feature orderings (see Appendix A.3).

We first study the models' performance in the zero-shot setting in Figure 5 (left). While performance generally improves with increasing depth, some models, such as *TabICL(v1)*, exhibit relatively lower accuracy in the early layers, suggesting that they may struggle to learn discriminative features effectively. We also report the EvR metric in Figure 5 (right), showing that a consistent trend emerges across models: early layers exhibit higher EvR, indicating a stronger reliance on exemplar-based generalization, while deeper layers show decreasing EvR, reflecting a shift toward rule-based reasoning. This transition suggests that TFMs progressively refine their representations, moving from surface-level similarity matching to more abstract feature utilization. Notably, the rate and extent of this transition vary across architectures, with some models maintaining relatively higher EvR even in deeper layers, indicating a persistent dependence on exemplars. Overall, these results highlight that generalization behavior is not static but evolves throughout the inference process, and that early-exit strategies may significantly affect the type of generalization exhibited by the model.

> **Takeaway 3.** Generalization shifts from exemplar-based to rule-based across layers.

## 5. Discussion

Our findings provide a complementary view to aggregate performance metrics to study tabular foundation models (TFMs) and provide initial results showing that models with comparable accuracy may rely on fundamentally different generalization strategies. The observed transition from exemplar-based behavior in early layers to rule-based reasoning in deeper layers suggests that TFMs internally refine their representations, progressively abstracting away from surface-level similarities. This has important implications for reliability: exemplar-based strategies may perform well in low-data regimes but remain sensitive to spurious correlations, while rule-based generalization can support more robust extrapolation under distribution shifts. Additionally, the dependence of generalization behavior on factors such as dataset size and model architecture indicates that inductive biases vary significantly across TFMs. These results emphasize the need for evaluation frameworks that probe internal mechanisms rather than relying solely on predictive performance, and they point toward future work on controlling and guiding generalization behavior in foundation models for tabular data.

**Limitations.** Our study has several limitations. We focus on synthetic binary classification tasks, which provide controlled settings but may not capture the complexity of real-world tabular data. Extending this analysis to more complex settings and regression tasks is an important direction for future work. Furthermore, adapting real-world datasets to enable similar controlled evaluations remains an open challenge and a promising avenue for further investigation. In addition, there are other generalization mechanisms that we do not explicitly test. Finally, we do not interpret absolute EvR values (e.g., what EvR = 0.1 vs. 0.2 implies quantitatively).

## Acknowledgments

This research has been funded by the Federal Ministry of Research, Technology and Space of Germany and the state of North Rhine-Westphalia as part of the Lamarr Institute for Machine Learning and Artificial Intelligence. Additionally, part of this research utilized compute resources at the Tübingen Machine Learning Cloud, DFG FKZ INST 37/1057-1 FUGG. A. Balef and M. Koshil also thank the International Max Planck Research School for Intelligent Systems (IMPRS-IS).

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

# A. Rule-/exemplar-based

## A.1. Experimental Setup Details

Following Dasgupta et al. (2022), we consider a 2D binary classification task with orthogonal feature dimensions and a simple generative model to distinguish predictive from distractor features. Features $x \in \mathbb{R}^2$ are generated as

$$x \mid z_{\text{disc}}, z_{\text{dist}} \sim \mathcal{N}(\mu, 1.0), \quad \mu = \alpha \begin{bmatrix} 2z_{\text{disc}} - 1 \\ 2z_{\text{dist}} - 1 \end{bmatrix},$$

with $z_{\text{disc}}, z_{\text{dist}} \in \{0, 1\}$ and $\alpha = 3$. Here, $z_{\text{disc}}$ determines the class label $y = z_{\text{disc}}$, while $z_{\text{dist}}$ is a distractor. The test set is the held-out group with $z_{\text{disc}} = z_{\text{dist}} = 1$. Notably, each class contains $N$ samples, and we use 100 repetitions for $N < 100$ and 10 repetitions for $N \geq 100$.

## A.2. Layers with Reliable Early-Exit Performance

We consider only layers for which models produce reliable outputs based on early-exit performance, defined as achieving at least $90\%$ of the final performance of the corresponding baseline model. This performance is evaluated as the average across the selected datasets from *TabArena* and *PMLBmini*, following the methodology of (Balef et al., 2026). The allowed layers for each model are summarized in Table A.1.

*Table A.1.* Layers used for analysis based on early-exit reliability.

| Model | Layers | |
|---|---|---|
| *TabPFN(v2)* | 5–12 | |
| *TabPFN(2.5)* | 7–24 | |
| *TabICL(v1)* | 1–12 | (only ICL predictor part) |
| *TabICL(v2)* | 1–12 | (only ICL predictor part) |
| *LimiX-2M* | 2–12 | |
| *LimiX-16M* | 4–12 | |

## A.3. Effect of the number of training samples

Here, we provide additional details about the experiments. We also report the feature-level bias (FLB), defined as follows.

Following Dasgupta et al. (2022), we study feature-level bias (FLB) using cue-conflict (CC) probes, which capture a model's preference for specific features during learning. In the CC setup, as shown in Figure A.1, predictive cues are deliberately put in conflict, allowing us to reveal which feature a model prioritizes when cues disagree, thereby exposing feature-level preferences. FLB quantifies the deviation from chance accuracy in the CC condition, where positive values indicate reliance on the discriminative feature rather than the distraction feature. Formally, for a classifier family $\mathcal{F}$, these metrics are defined as

$$\text{FLB}(\mathcal{F}) = \mathbb{E}_x[A(y, f^{cc}(x))] - 0.5,$$

where $f^* \in \mathcal{F}$ denotes a classifier trained on the corresponding training set, and $x$ is drawn from the test set.

For the experiments in the main paper, we report the average performance across different perturbation settings:

- **No Perturbations.** Baseline setting without any modifications to labels or feature ordering.

- **Label Flipping Effects.** Evaluates whether models are sensitive to label flipping perturbations.

- **Feature Swapping Effects.** Examines how models utilize distractor features and whether feature order influences performance.

- **Combined Perturbations.** Measures the joint effect of applying both label flipping and feature swapping simultaneously.

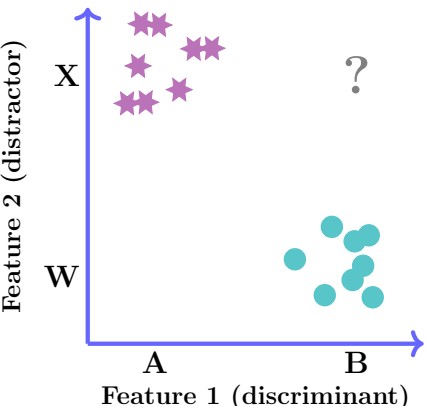

*Figure A.1.* Cue-conflict evaluation setup. The model is tested on a held-out combination (`BX`) while trained on `AX` and `BW`. Performance on the held-out quadrant directly measures feature-level bias, revealing whether the model prioritizes feature identity or feature value when cues conflict.

Averaging across these perturbation settings ensures that the reported results are not biased by a particular label ordering or feature ordering configuration.

Figure A.2 shows the EvR and FLB under the perturbations. As shown, the overall trends are consistent across models; however, for *TabICL(v1)* and *TabICL(v2)*, we observe that changing the label or feature order can noticeably affect EvR.

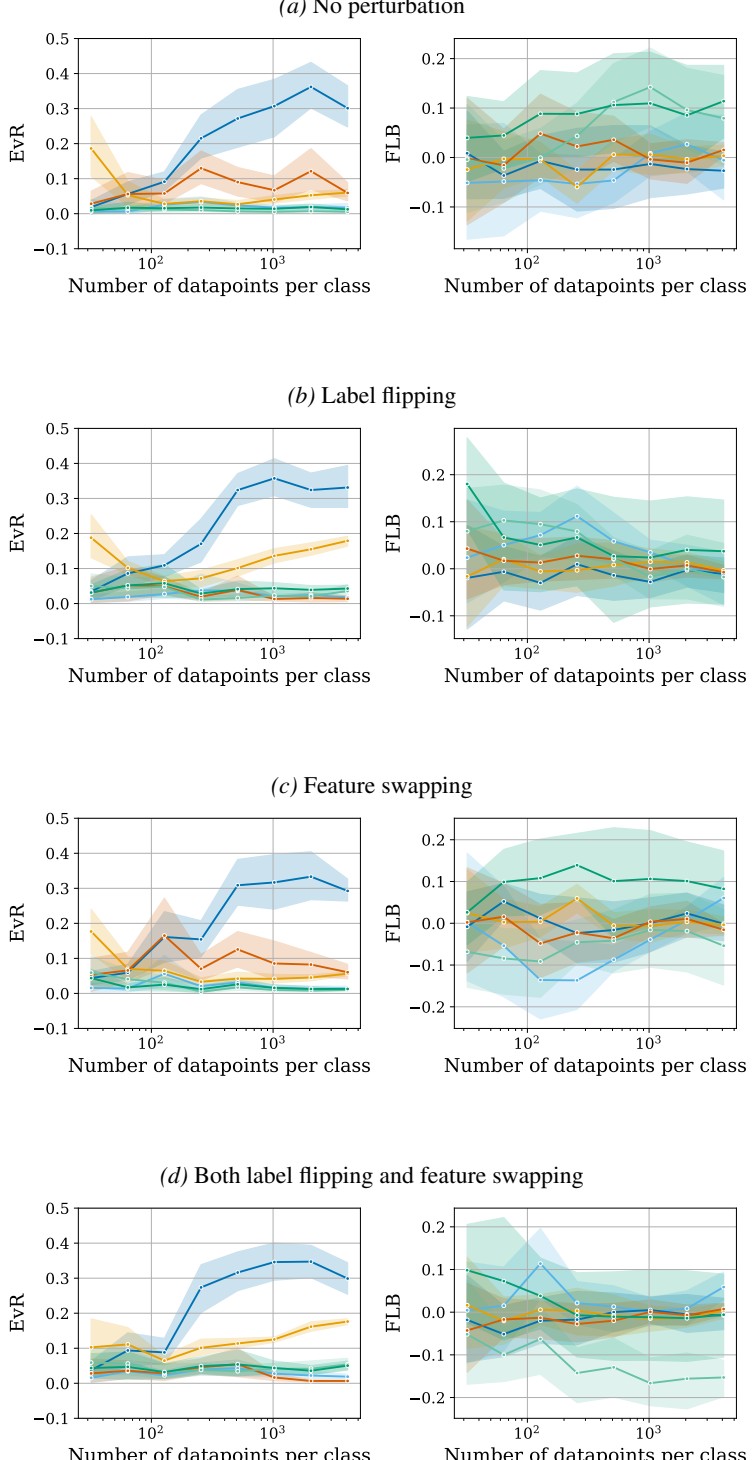

*Figure A.2.* Analysis of exemplar versus rule-based generalization (EvR, left) and feature-level bias (FLB, right) under four settings: no perturbation (top), label flipping (second), feature swapping (third), and both transformations (bottom).

## A.4. How does EvR evolve across the internal layers of TFMs?

Figure A.3 shows how the EvR, zero-shot accuracy, and partial exposure accuracy evolve across the layers under all four perturbations. As shown, the overall trends are consistent across models.

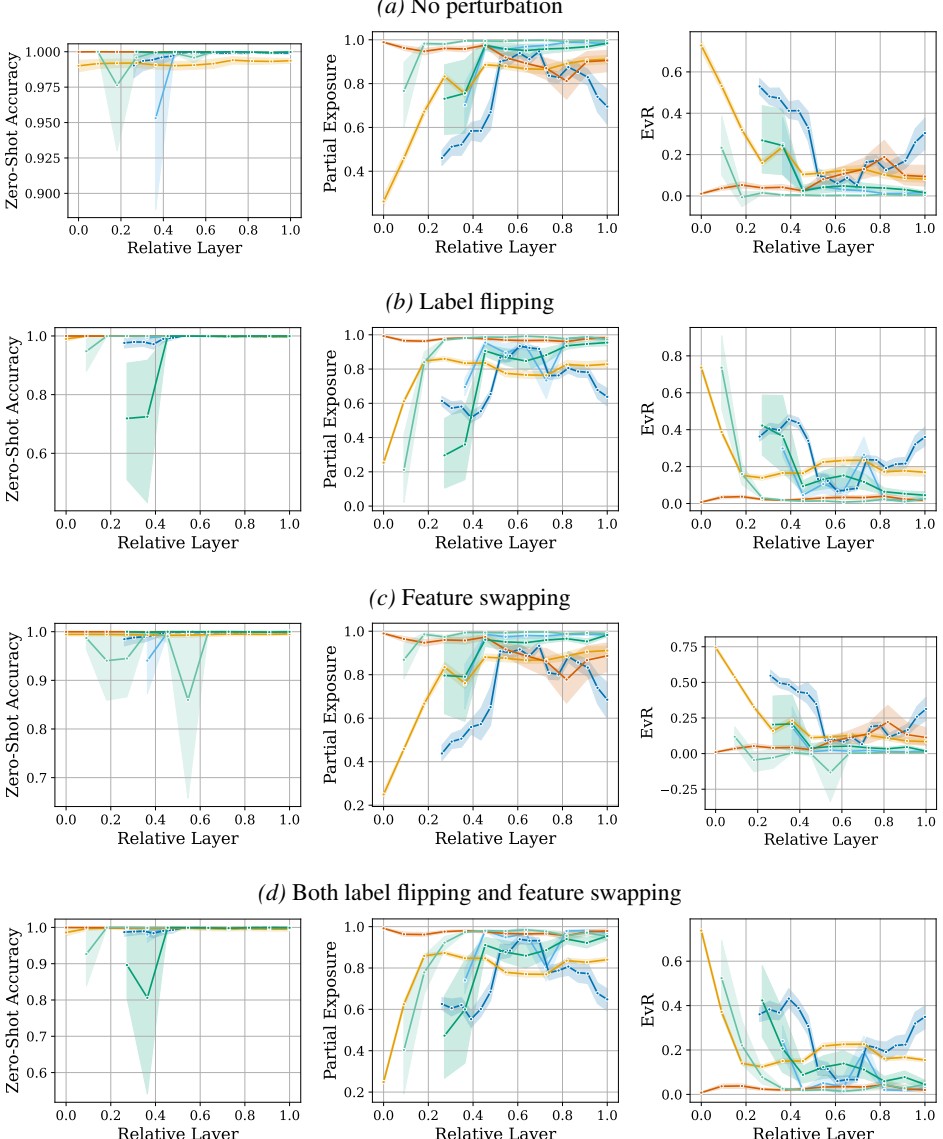

*Figure A.3.* The changes in EvR, zero-shot accuracy, and partial exposure accuracy across layers under all four perturbation settings: no perturbation (top), label flipping (second), feature swapping (third), and both transformations (bottom).

