# OpenReview forum: "Do Tabular Foundation Models Learn Rules or Memorize Exemplars?"
_ICML.cc/2026/Workshop/FMSD — FMSD @ ICML 2026 Poster_

### Official Review · Reviewer_2rUS · 2026-05-12
**Do Tabular Foundation Models Learn Rules or Memorize Exemplars?**

**Rating:** 5
**Confidence:** 3

**Review:**

## Summary
The author investigates the problem of inspecting the inference behavior of existing tabular foundation models. Specifically, Exemplar-vs-Rule (EvR) is examined across models, and the author concludes that existing tabular foundation models reveal both rule-based and exemplar based generalization, and the representation of intermediate layers demonstrate that earlier layers are more exemplar based and deeper layers are more rule-based.
## Strength
1) The problem being investigated is interesting, and it is very important in terms of enhancing researchers’ understanding of the behavior of existing tabular foundation models.

2) The method and experimental setup are well written.
## Areas for Improvement
1) The testing data, though is straightforward to be displayed as a paradigm scenario, having one or a few real world datasets would strengthen the claim of the observed model behavior.

2) The metric of EvR is questionable. By quantifying the difference between zero-shot and partial-shot, most of the time such metrics tell us about how good the model is in terms of adapting to new scenarios rather than indicating whether the model is relying on exemplar or generalizable rules. This also leads to the conclusion of the paper being questionable.

3) It is well known that for deep neural networks, deeper layers extract more abstract features. Earlier layers tend to focus more on raw pattern extraction, such as identifying fixed sets of filters, extracting local information, etc. Moreover, despite the issue within the chosen metric, many of the time, whether the model tends to be more exemplar based or more rule-based depends on the need of the application scenario. Considering these factors, a better experimental design is expected to enhance the convincingness and value to the community.

## Detailed Comments
The detailed suggestion is stated in section *Areas for Improvement*.
## Justification of Scores
Although the topic of the work aligns with the theme of the workshop titled “Foundation Model for Structured Data”, the experimental design is questionable, and the conclusion being derived is incremental.

---

### Official Review · Reviewer_wGEf · 2026-05-13
**Study on Generalization of TFM**

**Rating:** 7
**Confidence:** 3

**Review:**

Summary:
The paper investigates a fundamental question about tabular foundation models (TFMs): when these models make predictions through in-context learning, are they actually discovering abstract predictive rules, or are they mostly relying on memorizing and comparing against the examples provided in the context window?


Strengths:
The paper’s key strengths are primarily methodological rather than purely empirical. Its biggest contribution is that it moves beyond standard benchmark accuracy and directly probes how tabular foundation models generalize. This is important because most prior work on TFMs focuses on whether they perform well, while this paper asks what internal strategy they use to achieve that performance. That shift from performance evaluation to mechanistic analysis is a major strength.


Areas for Improvement:
Chat.openai.comThe paper could be improved by strengthening the connection between the synthetic setup and real tabular tasks. Right now, the experiments are clean and useful, but they are mostly based on controlled binary classification problems. That makes the mechanism easier to isolate, but it leaves open whether the same rule-versus-exemplar behavior appears on messy real-world datasets.


Detailed Comments
The current evaluation is almost entirely based on synthetic controlled tasks. While this is appropriate for isolating mechanisms, the paper would benefit significantly from testing whether the observed behaviors persist on real-world tabular datasets with natural spurious correlations, feature redundancy, and domain shift. For example, can the authors design semi-controlled experiments on datasets from healthcare or finance where distractor features are artificially introduced? This would help bridge the gap between mechanistic insight and practical deployment.


Justification of Score:
I would recommend acceptance because the paper asks an important question that standard TFM benchmarks largely miss: not whether tabular foundation models perform well, but why they perform well and what kind of generalization they rely on. The main strength is that the paper provides a clean, controlled way to separate rule-based behavior from exemplar-based behavior. This is valuable because high accuracy alone can hide very different mechanisms. The finding that different TFMs with similar performance can rely on different strategies is both practically important and scientifically interesting.

---

### Official Review · Reviewer_frjP · 2026-05-21
**Insightful analysis of TFM generalization strategies with minor presentation flaws**

**Rating:** 6
**Confidence:** 4

**Review:**

This paper investigates how Tabular Foundation Models (TFM) generalize during in-context learning (ICL), specifically whether they learn underlying abstract rules or simply attend to similar examples in the context.
The authors adopt a controlled 2D binary classification framework from prior work where the model is evaluated with and without distractor features (which make some of the test points closer to points from the wrong class). They compute an Exemplar-vs-Rule (EvR) metric which compares the two evaluations for size SOTA TFMs, and observe how the performance varies relative to sample size, feature counts and architectural layer depth.

Strengths:

1. The paper addresses a clear diagnostic research question which is important to the TFM community.
2. The experimental setup is sound and interesting. The results can be clearly viewed on  2D images.

Areas for Improvement:
1. The experiments do not lead to very clear insights. For most experiments different models behave differently and therefore the "take-home message"  is not very strong.
2.  The paper completely omits a discussion of the evaluated models' original pre-training data configurations. Since these TFMs are pre-trained on entirely separate synthetic prior distributions, the variation in their downstream EvR metrics may simply reflect differences in their pre-training data mixtures. For instance, the number of distractor features or the frequency of strict rule-based functions present during pre-training likely affects their final generalization biases. Connecting the models' underlying pre-training histories to their downstream EvR profiles would provide much-needed clarity. This can be shown using small scale models with full pre-training experiments.
3. Some of the experiment description is inconsistent. See details below.

Detailed Comments:
1. There is an inconsistency regarding the training data of the "partial exposure" setup. Figure 2  includes quadrant BW and the text does not (lines 89-90: " he model is given data from quadrant AW and AX"). I assume the text is a typo.
2. What is the role of the white points in figure 3? The full colors suggest that the model is tested on a dense grid of test points and not only  the white points in quadrant BX.

Justification for Score:
The paper formulates a reasonable research question and executes a sound empirical setup.  While the insights are not very significant, they are still valuable to the community and will contribute to the discussions at the workshop.